# Serious Illness Conversations in Pediatrics: A Case Review

**DOI:** 10.3390/children7080102

**Published:** 2020-08-18

**Authors:** Camara van Breemen, Jennifer Johnston, Matthew Carwana, Peter Louie

**Affiliations:** 1Canuck Place Children’s Hospice, Vancouver, BC V6J 2T2, Canada; plouie@cw.bc.ca; 2Two Worlds Cancer Collaboration Foundation, Kelowna, BC VIY 1G7, Canada; 3Faculty of Medicine, University of British Columbia, Vancouver, BC V6T 1Z4, Canada; Matthew.Carwana@cw.bc.ca; 4Chilliwack School District, Chilliwack, BC V2R 2E6, Canada; bradjen@shaw.ca; 5British Columbia Children’s Hospital, Vancouver, BC V6H 3N1, Canada

**Keywords:** childhood cancer, communication, palliative care

## Abstract

The Serious Illness Conversation Guide program developed by Ariadne Labs, a Joint Center for Health Systems Innovation, includes a list of patient-centered questions designed to assist clinicians to gain a more thorough understanding of their patient’s life in order to inform future care decisions. In July 2017, specialist pediatric palliative care clinicians at Canuck Place Children’s Hospice (CPCH) (Vancouver, BC, Canada), adapted the original guide to use with parents of children with serious illness. This tool is referred to as the Serious Illness Conversation Guide-Peds (SICG-Peds). Using the SICG-Peds, along with enhanced communication skills, can help illuminate the parents’ (child’s) understanding of illness and the values they hold. Expanding the application of the guide will promote goal-based, efficient, comprehensive and consistent communication between families and clinicians and help ensure that seriously ill children receive care that is tailored to their needs through the disease trajectory. This paper explores the guide through the lens of a case study. The steps—seeking permission, assessing understanding, sharing prognosis and exploring key topics (hopes, fears, strengths, critical abilities and trade-offs)—as well as formulating clinician recommendations, are described.

## 1. Case Study

Carter, an 11-year-old boy, was diagnosed with osteosarcoma, isolated to the left femur, in November 2016. He received standard osteosarcoma chemotherapy with cisplatin, methotrexate and doxorubicin. He also had surgery to remove the tumour and a total femur replacement. Pathology from the resection showed that the tumour had responded poorly to the initial chemotherapy (30–40% necrosis and residual viable tumour). Carter finished chemotherapy in June 2017 and was free of disease. However, imaging performed 3 months later showed that the tumour had already recurred in both lungs and in several spots in his bones.

In light of his poor prognosis, the managing oncology team made a referral to the pediatric palliative care team for current and future symptom management and to provide ongoing communication and family support. Carter’s parents Jen and Brad decided that their priority was to spend maximum quality time at home together. Hospital visits for chemotherapy were declined, but Carter was enrolled in a phase 1 study of a new oral chemotherapy combination.

Within 2 months, Carter experienced increasing pain in his back and right leg as well as some shortness of breath. He received palliative radiation to lesions in his pelvis, leg and lungs with the goal of decreasing symptoms in time for his Wish trip to Disneyworld. Carter and his family travelled for the Wish trip. However, after their return, his pain and symptoms of dyspnea increased. He was admitted to the hospice setting for infusion of opioids to control his pain and for his parents to receive support. It was clear that Carter wished to return home as soon as possible. The care team’s goal was to ensure that he and his parents had an understanding of what to anticipate as Carter’s disease progressed.

Over time, Carter experienced extremely complex and intense symptoms of pain, dyspnea, and changes in function. Although Carter’s pain was initially managed with oral and transdermal medications, he eventually required two continuous infusions through his central line for intense pain. This was managed by pediatric palliative care nurses and his parents. The pediatric palliative care team provided care to Carter over 17 months, initially as consultants and then as primary providers. Carter had several outpatient consults, two pain- and symptom-management stays at the hospice (11 and 5 days) and care at home with 42 nursing visits and regular scheduled telephone contact. His parents had access to 24 h phone support and urgent response visits. Family and counseling support was also provided by the interprofessional team to parents and siblings through in-person contact and via telephone. Carter died in his home with his family and his beloved dog present on 16 March 2018. The family continues to receive bereavement support.

Over the illness trajectory, the care team grew to know Carter, his siblings (Jack, Marley and Mason) and his parents, Jen and Brad, through relational practice and use of dynamic and expert communication. The pediatric version of the Serious Illness Conversation Guide was used formally in several family team meetings by a pediatrician trained in the use of the guide. A nurse practitioner, nurses and counselors also engaged in ongoing conversations that utilized core aspects of the guide in as-needed conversations within the home setting and by phone. Through this collaborative communication, parent and family values, wishes and hopes were continually explored to provide thoughtful planning and support.

## 2. Introduction

Meaningful conversations with seriously ill patients are characterized by gaining an understanding of patients’ goals, sharing prognosis along the trajectory of illness, and responding skillfully and sensitively to emotions [1,2,3,4]. Pediatric advance care planning should include a family-centered approach that considers hope and non-medical concerns alongside honest and practical information that allows parents to actively participate in decision making and prepare for end-of-life circumstances [5,6,7,8]. Collaborative communication and clinician skill in providing informed recommendations based on parent (child) values requires competencies that can be difficult to define. However, a framework for addressing standard questions and key topics in serious illness can be a helpful tool that enables consistent communication among different healthcare professionals and parents (child) so all concerns are addressed on an ongoing basis.

The Serious Illness Conversation Guide (SICG), developed by Ariadne Labs, a Joint Center for Health Systems Innovation (Boston, MA, USA) is a structured communication tool that provides clinicians with psychologically informed language to assess illness understanding and patient information preferences; share prognosis according to patient preferences; explore patient values, goals, and care preferences; and make recommendations based on patient priorities [1,9]. The SICG is unique in that it is designed to promote conversations that focus on what is most important to patients living with serious illness and to explore factors that are key to maximizing quality of life [1]. Though physicians and nurse practitioners typically provide information on prognosis, the other questions and sections of the guide can support communication by nurses, social workers, residents/trainees, and counselors. In July 2017, specialist pediatric palliative care clinicians at Canuck Place Children’s Hospice (CPCH) adapted the adult SICG process to use with parents of children with serious illness [10]. It was essential to adapt the guide in order to incorporate family-centered language, acknowledge that parents are the primary decision makers, allow for flexibility around a child’s developmental stage, and consider the difficulty of prognostication in children with rare diseases. The adapted guide is referred to as the SICG-Peds (Table A1 and Figure A1) and is now a two-page document. To date, CPCH has trained over 200 pediatric physicians, nurses, nurse practitioners, social workers and other allied professionals in the use of the guide in Vancouver, Ottawa, and Montreal, Canada and Hyderabad, India.

This paper explores the SICG-Peds through the lens of a case study, with a particular focus on how the members of the oncology and palliative care teams communicated with each other and the family to provide optimal care at hospital, in hospice and at home. The specific aspects of the guide—setting up the conversation, assessing understanding and information preferences, sharing prognosis and exploring key topics (hopes, fears, strengths, critical abilities and trade-offs)—as well as formulating clinician recommendations, are described. It is important to note that although the guide can be used in its entirety during a consultation and at various stages of the illness, we have found that the guide can and should be used in a more informal way, in the moment, at the bedside, after a bad night or when new fears emerge or alternate hopes are explored. Again, this is performed by those clinicians at the forefront of care and in the moments where a connection of common understanding between clinician and parent and child should be explored before an action is taken. The case has been used with permission of the family and is co-authored by a family member.

## 3. Components of the SICG-Peds

### 3.1. Setting up the Conversation

For various reasons (e.g., busy schedules, multiple teams, or the power imbalance often inherent between parent and clinician), clinicians may neglect to prepare parents for a meeting to discuss their child’s illness and prepare for future decisions. Seeking permission and collaborating on a time to meet allow families to prepare their questions and to request the attendance of those who support them. It can also alert parents that serious news will be discussed. This action models respect and is the first step in a collaborative partnership that emphasizes the relationship between clinicians and families to build common understanding, albeit through differing perspectives [11].

Clinicians are often unsure how to introduce difficult communication about poor prognosis and end-of-life issues [12]. Treating specialists may also hesitate to refer families to palliative care despite knowing that this service can be beneficial [13]. For Carter’s family, their oncologist had affirmed the parents’ worries that the clinical symptoms may indicate that the disease had progressed. He asked the parents’ permission to introduce the CPCH team to assist with care, particularly with pain and symptom management, care planning and issues related to communication.

### 3.2. Assessing Understanding

When first meeting with parents, it is crucial to gauge their understanding of how they feel their child is doing, particularly in the face of difficult news or changes in their child’s functional status. This goes beyond acknowledgement of the ‘knowing’ parents have in regard to their child and their condition; it sets the tone and leads into how the parents are feeling, what their concerns are, how accurate their information is (congruency between parents and clinicians), and what words they use to describe the illness and symptoms. Jen and Brad, like many parents, knew what increasing pain signified in the context of a bone cancer diagnosis. They had integrated the information provided by the oncologist at the time of diagnosis and then again at relapse. They understood the seriousness of the diagnosis and were diligent in tracking their son’s response to treatment, level or change in activity, and symptoms. While always hoping that the cancer would be cured, they could voice their greatest fears of worsening disease through explicit assessment of the past several weeks and months.

Assessing understanding is a dynamic and ongoing process in collaborative communication. This can be performed in a formal team meeting at times of investigations (scans, blood work, etc.) or when symptoms arise. Parents’ understanding of the overall nature of the condition—‘yes I know his cancer is incurable’—is augmented by inquiries about day-to-day understanding. For example, gaining a perspective of why the pain is worse at night may lead to enhanced understanding of the link between periods of quiet and increased anxiety. Furthermore, parents who have not cared for a seriously ill person before may misunderstand symptoms or attribute meaning that is not accurate. Parents will be wondering, ‘Am I missing something or will I know when the end is near?’ Clinicians who take time to explore through questions and reflections will gain valuable information about ways to communicate what is happening now and what may happen in the future.

### 3.3. Assessing Preference for Information

It is important to understand how much information both the parents and child want to know and how they want to receive updates on medical issues. When asked about how much information they would like to receive, Jen and Brad stated they wished to know everything about Carter’s current medical status and prognosis as it became available. They shared that they wanted to be able to prepare themselves and their other children for what was to come. They acknowledged that they wanted guidance in how to support Carter in his understanding while remaining sensitive enough not to overwhelm him. They feared information may increase his anxiety and they were unsure how to support him if he were to lose hope. With the help of the guide in assessing a child’s understanding and involvement (Figure A1, top left), a conversation with Carter about what he understood about his illness ensued. He shared with the nurse practitioner that he knew the cancer was growing based on the changes in his body and that he knew the treatment was not working. He said he did not need more information and that talking about the future in regard to his cancer was stressful and increased his anxiety. He voiced his trust in his parents to make good decisions for him. Rather than making his own decision in terms of treatment, he wanted choices about where his care would be. He was clear that home was best, hospital was worst and the hospice only if he had to be there to help his pain. He knew these choices meant that it would limit the intravenous chemotherapy treatment and his parents and oncologist supported him in this decision. Though not discussed with Carter, this also meant it limited critical care interventions or investigations that could only be performed if he were in hospital. Carter decided, and this was documented and communicated to the team, that he wanted to know the general plan of the day (“we are making plans today so we can go home tomorrow”) and to be given choices about the type and manner of care given (“Do you want a shower in the morning or evening?” or “How do you want to lie when we change your dressing?”). As time progressed, with each functional change (decreased mobility, bowel and bladder changes) or increase in pain, his quiet worry and stress surfaced. He relied on his parents and the nurses to affirm that he was safe and that there was a plan to help manage the physical and emotional difficulties. As Carter weakened and spent much time in bed near the end of life, it was clear that words mattered little but that the physical closeness of one or both parents at all times was what he relied on to feel comforted.

### 3.4. Prognosis

Prognosis, framed as the amount of time one has left to live, is an extremely unreliable landscape. Although clinicians’ prognosis tends to become more accurate as death approaches, the most experienced palliative care experts remain in awe of the body and the elusiveness of knowing when the end will be. Giving ‘time’ and being wrong can disrupt the trust relationship with parents.

However, parents often request the discussion of time (couched with the acknowledgement of uncertainty) so they can plan and make decisions about how to spend the time remaining. Importantly, one study reported that clinicians know, on average, 100 days prior to parents that the child is going to die. When physicians and parents have the shared recognition that time may be short, there can be a stronger emphasis on treatment directed at lessening suffering and greater integration of palliative care [14]. Prognosis sharing, however, is much more than divulging the expected length of time remaining. Prognosis is about predicting what is likely to happen in the future while giving guidance and support about care that will help their child and interventions that will not [15]. It involves sharing information and then responding to the emotional and practical questions and distress that arise as parents contemplate their child’s deterioration, potential suffering, death and life without them.

Experts describe giving short, concise, and honest information to parents [5,7] and the SICG references words that patients themselves have found helpful, including the use of time, uncertainty or function prognosis [9]. When adapting the guide for use in pediatrics, CPCH staff added two categories of children’s health to the framework to aid in prognosis wording: the clinician’s assessment of fragility and stability (Figure A1, top middle). Fragility assesses the likelihood that a child will experience a sudden deterioration in a major body system. Stability encompasses the current care needs and the rate of change in the child’s wellbeing in terms of symptom burden, care need, functional, or developmental decline.

When the palliative team first met Carter and his parents, he had a great deal of pain in his leg and pelvis that was affecting his whole person; his appetite was reduced, his mobility was compromised, and his quality of life was poor. However, the metastases in his lungs were small and he was not having any problems with respiratory or cardiovascular health. Although he was thin, in pain, and very uncomfortable, his parents could be reassured that his breathing was stable. If his pain could be better managed, the hope was that his appetite would improve and he could be more active. Using the framework above, Carter was in a period of low fragility, but also low stability. This assessment signaled to the team that medical directives regarding critical events could wait and that a return home once symptoms were managed was a reasonable goal.

Before he was discharged home, nurses taught parents and Carter how to track his health and how symptoms or functional changes should be managed. There was also exploration of when the parent should call. When exploring the key topic, fear, in the guide, Carter’s parents identified that unremitting pain or difficulty breathing was a worry. As these symptoms were both quite likely to occur in the future, the team could prioritize making management plans for these symptoms, and the parents were assured that they had the 24 h contact line to the nurses and the doctors of CPCH. Symptom management orders were given to parents and copied to the chart so the care team could reference if the parents called. It was not necessary at this point to write out advance directives related to Do Not Resuscitate (DNR) as Carter was relatively stable and robust and a sudden event was not likely. The balance of preparing parents while not overly burdening them with unlikely scenarios or going too far into the future must always be considered when discussing advance directives.

As the cancer progressed and pain and dyspnea worsened, it became apparent that Carter’s fragility was increasing and that a terminal event could occur. This change necessitated a prognosis conversation that again explored parents’ understanding, goals and fears along with recommendations of what types of interventions would benefit Carter and what would not. Advance directives, including a DNR (along with all the interventions that would be of benefit, e.g. analgesia and oxygen), were documented. Care at home was again affirmed as being the first choice. Additional support for urgent response (24/7 home nursing visits) was also added as significant changes were becoming more likely. The limits of specific critical interventions were discussed and documented approximately 4 weeks prior to his death, at the time where both clinicians and parents could see and clearly acknowledge end of life was near and calling emergency services would not be of benefit. 

### 3.5. Exploring Key Topics

#### 3.5.1. Hope

There is a frequent misconception that parents will lose hope if they are provided with a prognosis and encouraged to initiate advance care planning; indeed, parents value and rely on honest information [5]. Parents remain hopeful even in the face of understanding that their child has a serious illness and sharing information does not diminish this hope [15]. Clinicians should respond to emotion through deep listening. The guide advises the clinician to use the wish, worry, wonder framework as a way to align with the parents’ hope, acknowledge concerns, and then propose a way to move forward (Figure A1, middle).

Throughout Carter’s treatment, the hope that a cure could happen prevailed. At the time of relapse, there was hope that the radiation would improve pain and give him more function. This was followed by hope that the phase 1 drug would improve quality of life and extend time. Hope changed to focus on time with family and trying competitive wheelchair basketball and returning to school. Jen and Brad voiced that ultimately their hope was that Carter would not be afraid or endure pain or suffering. If there was pain, they hoped they would know how to help him. Hopes were defined through small and large ways. A nurse who cared for Carter in his home learned that Carter loved movies and he was aware that the new movie Black Panther was being released. This conversation arose naturally from a nurse eager to engage Carter in things that he liked and made him happy. Carter expressed his sadness and seemed resigned that he likely would not be able to attend a movie given his functional losses, significant symptoms, fatigue and his private nature (he could not imagine going to a crowded theatre). The nurse reflected his sadness and explored other losses that were hard given his cancer. Through more conversations, and with planning from the interprofessional hospice team, a private viewing of the film in a theatre close to their home was proposed to Carter and his family. Transportation, nursing and symptom support planning was initiated. Carter was able to invite as many friends and family as he wanted. This event occurred just 15 days prior to his death.

#### 3.5.2. Fears

Asking parents to contemplate their fears and worries about their child’s future can be difficult territory to navigate. Parents’ role to provide for and protect is universal and the contemplation of their child dying before them goes against the natural order. It can be distressing to clinicians to hear such raw emotions. However, exploration of fears and worries will help clinicians gain insight into what parents and the child are thinking and equip them with a foundation to build in support. Fears about anticipated symptoms, the dying process, family functioning, and decisional regret among others can consume and deplete parents. When specific concerns are expressed, they should not be considered problems to solve or to placate but rather a space to provide deep listening and reflection of the sadness, anger, or anxiety that those thoughts bring. Jen and Brad, like most parents, feared the suffering their child might endure physically, emotionally and spiritually as the symptoms became worse. They worried for their other children and how they would cope with witnessing their brother’s dying process and death. In these often difficult conversations, clinicians learn of the burden the parents carry and thus can align, guide and perhaps even lessen the load through support and planning. Though no one can ever be assured that a person will not suffer, mitigating suffering through expert palliative care can greatly impact the illness course including guidance to parents on how to talk to their child, preparation for medical aspects surrounding end-of-life and sibling support during this difficult time [13]. Furthermore, detailed plans for the anticipated worries and a system that can respond quickly can ease the helplessness and isolation the parents are expressing. Pediatric palliative care expertise should be accessible 24 h/day. 

#### 3.5.3. Strengths

When asked about their strengths, Carter’s parents easily expressed that the connection they have with their children and wider family and friends was their greatest strength. It was clear that time together was enjoyed by all and gave each of them comfort and resilience. The time during treatment and the time at the hospice were difficult as they balanced the needs of Carter and their other children, which often required that they were apart, with parents trading off duties. Jen also spoke of Carter’s quiet strength. He could endure so much and complained so little. Carter was a good friend to others. He was an exceptional athlete and excelled at soccer and hockey prior to his surgery. He had a competitive spirit with good sportsmanship. His closest, long-term friends continued to be in his life even after attending school was no longer an option. Online and in-person gaming, visiting and spending time was very important to all of them. Carter also had an incredible relationship with his dog who seemed to understand just what Carter needed. The nurses who cared for Carter in the home spoke so fondly of his gentle strength and kindness to them. He always said thank you—no matter the size of the task or how hard it was on him (namely the dressing change to his coccyx). He asked questions and was interested in their lives and stories. He showed excitement for one of the nurses as she shared that she was taking her own children to Disneyland. Carter was a deeply gracious and kind boy and caring for him was a privilege.

#### 3.5.4. Critical Abilities

Although it is impossible to know what one can live without when projecting into the future, the question of critical abilities can have a key place in deep communication work with parents of a child with a serious illness or with the child. At times, it is not necessary to ask directly but for the expert clinician to listen to the underlying messages of what makes life worth living. This may include aspects of independence, the working of basic senses (sight, taste, touch, and hearing) and/or the ability to have meaningful engagement with others. Jen and Brad spoke about the abilities Carter had already lost due to the cancer treatment. The changes to his competitive sport outlets and his ability to give time to his academics were a great loss. Carter grieved his losses (as did his parents) and adapted to life in the months that preceded his death. He continued relationships with friends and family, though over time that circle became smaller matching the energy he had and intimacy he required. Good days changed from going out of the house to meet with friends, to getting out of bed and playing video games on the couch. As weakness increased and pain worsened, sleeping peacefully or being comfortable watching a movie was the goal. Parents, a sibling (or three) or his dog close by was a must.

#### 3.5.5. Trade-Offs

Trade-offs as elicited by “what are you willing to do if your child gets sicker for the possibility of gaining more time?” is a key topic that is complex and nuanced, particularly in the pediatric context where parents are usually the primary decision maker. Helping parents understand the harms, benefits, burdens and risks of any treatment or intervention is an ongoing dialogue that weighs helping them differentiate between what constitutes good care and care that may be multi-faceted but not holistic. The decision at the time of diagnosis to have surgery and chemotherapy for the chance of cure was readily made, even though it carried the burden of loss of the femur, hospitalizations for chemotherapy and supportive care and many outpatient visits. When radiation was presented as a palliative measure to mitigate symptoms and potentially restore function, the short course of extra appointments and the side effects incurred were also viewed as reasonable by the family. In considering further chemotherapy in light of the advancing disease, the parents, through discussion with Carter, felt in-patient hospitalizations or the burden of chemotherapy that would likely entail transfusion support was too much of a cost. Time at home and avoiding the side effects of chemotherapy that would not cure the cancer, as well as avoiding the stressful hospitalization, was the difficult direction of care that they chose.

Parents view trade-offs differently and for some, choosing comfort-focused treatment seems unacceptable in the face of other treatments directed at the underlying condition or with the critical care support that may give the child more days of life. Parents express a desire to ‘leave no stone unturned’ and worry that they will regret that they did not try hard enough. Again, clinicians need to stay with the dynamic process of weighing “what is” and “what could be”, given all the options of care. It is essential that planning is not framed as a dichotomy of “doing something” (treatment, escalation of care) as opposed to “doing nothing”. Leaning into a palliative approach means describing all the ways the child will be cared for—physically, emotionally, and spiritually—and the reminder that family will not be abandoned regardless of their decisions. It is holding hope while also framing and exploring with the family—“if time is short”—what it is that they and their child need, where they want to spend that time and how clinicians can ensure that suffering is minimized. 

### 3.6. Recommendations

As the conversation draws to a close, clinicians should provide a summary of the content of discussion and ensure that there is a sense of shared understanding. Parents want clinicians to display sensitivity, empathy and competency with them [16,17,18,19,20,21] and parents need to be prepared for what to expect from their child’s illness and potential treatments [22,23,24]. Recommendations need to correlate with the parents’ understanding and the values they express through discussion of key topics. Parents generally state a spectrum of wishes from requesting and hoping for all interventions and life-sustaining measures to understanding some treatments to be of benefit while others are comfort-focused. There can be strong congruency between parents’ values and clinicians’ assessments and there also can be significant differences. The job of the clinician is to hear the parents and to continue to work on and with them in relieving the tension and finding care and interventions that can be agreed upon. At each juncture of decision making, Carter’s parents made decisions that included consideration of the treatment options and their risks and benefits. Initially, the recommendations regarded standard treatment based on his tumour. Later recommendations included best treatment for his pain and symptom management. Recommendations also included responding to the parents’ requests for support for their other children and for Carter to be comfortable at home. Medical directives were described in what would benefit him and what would not. Ultimately this included the recommendation that when his breathing ceased and his heart stopped, holding him and caring for him was the best care that they could give.

## 4. Conclusions

Carter’s story highlights the way that the SICG-Peds can be used by pediatric palliative care clinicians to engage with families around sensitive topics. Parents with seriously ill children rely on clinicians’ ability to communicate well. Clinicians require competencies that include excellent listening skills along with the ability to delve into themes outlined in the SICG-Peds—seeking permission, assessing understanding, prognosis sharing, and exploration of key topics. This enables the parent–clinician connection to then create a path forward in caring for the child and the family that is centered on underlying knowledge, values, and support through the trajectory of the illness. As outlined, determining goals of care and decision making is not a one-off conversation. The use of the guide and components of the guide assist with revisiting understanding, hopes and fears within the context of the clinical picture shared through sensitive prognostications. Notably, parents who had the opportunity to engage in advance planning for their child reported that they felt more likely to be prepared for the child’s last day and better able to plan for their child’s location of death and rated their child’s quality of life during end of life care as good to excellent [25].

Clinicians at CPCH who have been trained and regularly use the guide have commented that the guide offers several advantages over solely relying on acquired communication skills. These include the ability to better listen knowing that one does not have to think of the next thing to say along with suggested wording for prognosis that can reduce the clinician’s anxiety in sharing information and provide a platform for common language between providers. Families can be better identified as needing a serious illness conversation as the team will consider early on ‘do we actually know the family’s values, hopes and worries?’ The guide also has the advantage of a format that is easily documented allowing a reference in which future conversations can be based, thus promoting consistency in discussions and messages across various members of the health care team.

A guided process can provide the foundation for excellent communication both within the team and with the family to improve the living and dying of the children and families that need palliative care. While this case highlights the use of the tool in a child with advanced cancer, it can be applied to families with children with a wide range of life-limiting or complex medical illnesses. The SICG-Peds is a tool that has been well received by both providers and families. It can be used to facilitate the meaningful conversations that contribute to high-quality, child and family-centered palliative care.

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
