# Peer review of "Serious Illness Conversations in Pediatrics: A Case Review"

_children, 2020, doi:10.3390/children7080102_

Round 1

Reviewer 1 Report

In this manuscript, the authors - including a parent of the specific case described and used as the premise for the model of serious illness conversations in pediatrics - present a well-done examination of potential pediatric use of the 'Serious Illness Conversation Guide' program developed by Ariadne Labs for use in adults. The chosen case, managed for over 1 year's time inclusive of the Palliative Care team's initial consultation, subsequent primary management, and into the bereavement period is incredibly well revealed to the reader in a most instructive manner.

The authors are all to be commended for presenting us with such an exemplary use of this novel pediatric application. It is certainly bound to find wide applicability in helping pediatrics patients and their families and pediatric clinicians.

Author Response

Thank you so much for your supportive comments.  Very much appreciated.  

Reviewer 2 Report

Thank you for the opportunity to review this interesting, important and well written manuscript on communication skills in the context of paediatric palliative care (PPC).

The authors explore a conversation guide, developed by Ariadne Labs, adapted by PPC clinicians of Canuck Place Children’s Hospice for its use in the context of parents with severely ill children. Based on a case review the paediatric version SICG-Peds is explored and explained.

In whole, I believe this is a valuable article for health care professional working in the field of severely ill children and their families.

There are some small comments and a concern related to the tool which cannot be easily accessed by the reader.

Abstract: References should not be included in the abstract (1).

Ref 1 – leads to a web page requiring login – I do not believe that many will undertake this procedure even if it is rather straight forward. On this page you need additional steps again to find the serious illness conversation guide.

Line 87 – SICG-Peds – should be written consistently (Peds/peds)

Ref 16 seems to be incomplete or lacking.

Section: "Recommendations": The first sentence should be revised or omitted as it is somehow redundant.

Unfortunately, I could not find the Appendix A and B and can thus not comment on it.

Author Response

Thank you so much for your detailed review.  I have changed the abstract and found a better reference for the adult guide that can be accessed more easily than from the website.  I corrected the other issues and am hopeful have completed the other recommendations.  I very much appreciate your time in this. 

Abstract: References should not be included in the abstract (1). -

I have changed this and included the reference in the main body

Ref 1 – leads to a web page requiring login –I do not believe that many will undertake this procedure even if it is rather straightforward. On this page you need additional steps again to find the serious illness conversation guide. -

I included an article with the full Serious illness adult guide for more easy reference that does not require a login.

Line 87 – SICG-Peds – should be written consistently (Peds/peds) -

I reviewed and checked the consistency through the paper (I did not see the error on line 87)

Ref 16 seems to be incomplete or lacking. -

I have deleted this reference as it was a note to myself and I have rechecked the references for accuracy.

Section: "Recommendations": The first sentence should be revised or omitted as it is somehow redundant. -

I have revised this this with the last submission.